# Role of Clay Substrate Molecular Interactions in Some Dairy Technology Applications

**DOI:** 10.3390/ijms25020808

**Published:** 2024-01-09

**Authors:** Abdelkrim Azzouz, Vasilica Alisa Arus, Nicoleta Platon

**Affiliations:** 1NanoQam, Department of Chemistry, University of Quebec, Montréal, QC H3C 3P8, Canada; 2Station Expérimentale des Procédés Pilotes Environnementaux (STEPPE), École de Technologie Supérieure, Montréal, QC H3C 1K3, Canada; 3Catalysis and Microporous Materials Laboratory, Vasile-Alecsandri University of Bacau, 600115 Bacău, Romania; arusalisa@yahoo.com (V.A.A.); nicoleta.platon@ub.ro (N.P.)

**Keywords:** fermentation, milk coagulation, clay materials, aflatoxin retention, lactic acid removal

## Abstract

The use of clay materials in dairy technology requires a multidisciplinary approach that allows correlating clay efficiency in the targeted application to its interactions with milk components. For profitability reasons, natural clays and clay minerals can be used as low-cost and harmless food-compatible materials for improving key processes such as fermentation and coagulation. Under chemical stability conditions, clay materials can act as adsorbents, since anionic clay minerals such as hydrotalcite already showed effectiveness in the continuous removal of lactic acid via in situ anion exchange during fermentation and ex situ regeneration by ozone. Raw and modified bentonites and smectites have also been used as adsorbents in aflatoxin retention and as acidic species in milk acidification and coagulation. Aflatoxins and organophilic milk components, particularly non-charged caseins around their isoelectric points, are expected to display high affinity towards high silica regions on the clay surface. Here, clay interactions with milk components are key factors that govern adsorption and surface physicochemical processes. Knowledge about these interactions and changes in clay behavior according to the pH and chemical composition of the liquid media and, more importantly, clay chemical stability is an essential requirement for understanding process improvements in dairy technology, both upstream and downstream of milk production. The present paper provides a comprehensive review with deep analysis and synthesis of the main findings of studies in this area. This may be greatly useful for mastering milk processing efficiency and envisaging new prospects in dairy technology.

## 1. Introduction

Milk is the earliest food for humans and mammals after birth, containing essential nutrients for body growth and development. Unless directly consumed as such, milk is usually processed via both traditional and up-to-date methods of fermentation and/or coagulation into more stable and preservable derivatives. Most traditional methods developed by humans for milk processing have been scaled up into industrial technologies (Figure 1). Knowledge advancement in biochemistry allowed implementing other processes such as pasteurization, lyophilization, concentration, degreasing, and others.

Milk derivatives such as creams, butter, cheeses, yogurt, kefir, and other lactic acid products (LAPs), cheeses, and others are usually intended for direct consumption [1]. However, they can also be further processed as raw materials for manufacturing other more added-value derivatives used in gastronomy, pharmacology, cosmetics or ice cream manufacturing technology. Indeed, buttermilk and whey are by-products of the production of butter and cheeses, respectively, and are currently used, on a large scale, as emulsifying agents in bakery technology, in the manufacture of baby food, etc.

Most of these derivatives are considered as highly nutritious dairy products, their consumption has been subject to more or less controversial issues related to increased risks to human health. Some of these issues deal with heart and artery clogging and others with diabetes, Alzheimer’s, and some cancers induced by saturated fat in milk. Other issues involve milk contamination by aflatoxins, lactose intolerance due to the decreasing capacity of humans with age to produce enzymes for sugar metabolization, and fairly contradictory research and surveys regarding the effect of calcium on bone health.

This has stimulated scientists to elaborate strategies in this regard, in addition to those already developed for milk production, preservation for storage, and large-scale distribution. As a common feature, these strategies target the removal or conversion of one or more milk components, e.g., by using clay materials as toxin adsorbents and/or buffering agents to overcome excessive pH decrease below 6.4 due to sub-acute ruminal acidosis (SARA). In this regard, the production of a higher quantity and quality of milk with controlled pH was reported for livestock fed total mixed rations (TMRs) with a maximum clay content of 2% tested [2]. This value was authorized by the European Union [3] for approximately 222 million dairy animals, including cattle (60%), buffalo (12.9%), goats (18.5 %), and sheep (8.6%). With a hypothetical daily feed consumption similar to that for cattle, at 50 pounds of dry matter (Food and Agriculture Organization of the United Nations, https://ourworldindata.org/grapher/milk-production-tonnes, 18 December 2023), a maximum global clay consumption may be estimated to be ca. 100 tons clay/day throughout the world.

So far, milk production has been improved by knowledge advancements about the basic principles that govern milk chemistry and biochemistry in correlation with its composition and origin. Nevertheless, the diversification of milk-based food products and their quality improvement can only be achieved through a deep understanding of the physicochemical interactions of additives towards milk components. For instance, the recent trend of using clay materials, which include clays and clay minerals, in dairy technology is justified by the need for sufficiently available natural and low-cost materials displaying suitable surface properties for improving the profitability of some processes in dairy technology, with an emphasis on livestock feeding. Clay materials (clays and clay minerals) are believed to be harmless, with acceptable relative chemical stability. Nowadays, these materials are now regarded as being ideal food-compatible materials for these purposes by many countries, more particularly in Europe. Indeed, 15 clay minerals have been designated through Regulation (EC) No. 1831/2003 as binders in the category of additives used in animal nutrition for controlling pH fluctuations and contamination by organic and inorganic impurities sources [3,4]. Livestock feeds were mostly enriched by bentonites and their deriving montmorillonites, alone or in mixtures with biocompatible compounds [5]. Since 2012, the authorized clay materials have been considered as being safe for consumption and non-genotoxic, non-skin irritants, and non-skin sensitizers, unless inhaled, since their silica content is a toxicity source for human and animal health [3]. Clay materials are also well-known to exhibit sharp edge lamellae or pointed ends according to their structures, which provoke eye irritation by direct contact and even skin abrasion upon abusive cosmetic massages.

Though milk physical chemistry and biochemistry are well established by scientists in this area, there is still a glaring lack of knowledge correlation between the structure, composition thermal/chemical stability with the colloidal behavior of clay materials in biochemical media. Cationic clay minerals are crystalline aluminosilicates with surfaces bearing negative charges with hydrophobic siloxy-rich silica islands and both out-of-plane and in-plane silanols in hydrophilic areas with p*K*_a_ values of 5.6 and 8.5 assumed to be close to those of pure silica [6,7,8,9]. This confers a colloidal and pH-dependent behavior of clay materials that determines the clay dispersion, particle size, and contact surface in aqueous media. In the pH range of milk and biochemical media, silanol deprotonation generates additional negative charges whose surface density decreases with decreasing pH up to clay coagulation–flocculation depending on the clay concentration and ionic force. Silica-rich clay minerals display lower colloidal properties with higher hydrophobic interaction and high affinity towards organic molecules, but less surface charge and exert less interaction with water being less influenced by pH fluctuations [10,11,12,13,14,15,16].

Fundamental knowledge for using clay materials in dairy technology should also include the very concepts of clays and clay minerals along with the heterogenous distribution of surface compositions that induces specific distribution of all types of interactions with dispersed species in aqueous media [7]. The challenging objective of this literature analysis resides in elucidating some interactions of milk components with solid surfaces and more particularly clay materials. For this purpose, a multidisciplinary approach is an essential requirement for matching adequate clay materials to a targeted process in dairy technology, more particularly in fermentation and coagulation. Providing in-depth knowledge on clay behavior in these processes is expected to open promising prospects for low-cost milk processing with weak consumption of lactic ferments and energy.

## 2. Solid Surface Contribution in Fermentation and Coagulation

Intensive research directions have been triggered in using clay materials as host matrices for enzyme immobilization and lactic bacteria growth, storage, and delivery in fermentation processes. Numerous works have also been devoted in the forms of research and review papers along with technical and scientific reports to the applications of clay as adsorbents for the capture of various undesirable chemical species upstream (aflatoxins) and downstream (fermentation inhibitors) of milk production. A synthetic analysis of these works shows that, on the one hand, in livestock feed, clay materials are ingested as such, and are supposed to be eliminated and loaded with aflatoxins and other drug excess, if any, without modification. On the other hand, in milk processing, these materials must be easily separated and recycled as well, if possible, without undergoing modifications. Conclusive results have been recently reported on the in situ capture of lactic acid (LA) by clay particles in the very fermentation broth [17]. Regeneration of the filtrated materials was achieved by ex situ ozonation after redispersion in a separate aqueous media [18]. Practical considerations require the use of removable bars, plates or permeable bags of clay materials in the form of grains or pellets.

Milk fermentation is an unavoidable natural degradation of fresh milk under inadequate preservation or storage conditions. Bacteria-driven fermentation mainly involves lactose oxidative conversion into lactic acid. A decrease in pH down to ca. 4.4–4.6 unavoidably triggers milk coagulation via the precipitation of caseins (α_s1_, α_s2_, β, κ), which account for ca. 80% of cow milk proteins, and many other processes [19,20]. However, coagulation, which involves milk separation into curd and whey, may also result from chemical processes in the presence of acidic species and/or enzymes. Unless extracted from the liquid media, the resulting coagulum still undergoes more or less pronounced proteolysis according to the clotting enzyme involved (Figure 2).

Fermentation is a complex bacteria-driven process with different physical–chemical reactions whose reaction pathway varies according to the bacteria strain. In time, this complex process has become a route for the production of a fairly wide variety of added-value LAP whose sensory properties vary according to the targeted fermentation and coagulation levels. For instance, yogurt preparation involves microbiological conversion of lactose and citrate. Simultaneous lipolysis of milkfat gives rise to most flavor compounds such as carbonyl compounds, alcohols, acids, esters, and other products [21]. Yogurt flavors also arise from the contribution of other products like LA, acetaldehyde, diacetyl, acetoin, acetone, and 2-butanone, which often disappear upon unavoidable lipid oxidative degradation into undesired aldehydes and fatty acids for prolonged preservation.

The main shortcoming of fermentative processes resides in the severe constraints in the selective control of lactic bacteria production and the unavoidable fluctuations in lactic bacteria activity that lead to pronounced and uncontrolled pH decreases. Excessively low pH is well-known to be detrimental to lactic bacteria, hindering their growth and activity. Milk coagulation can also be achieved through fermentation-free processes that use directly casein coagulating agents. This technique allows for overcoming this major drawback by avoiding lactose fermentation and the production of LA and other undesirable species.

Rigorous monitoring of the sensory properties of the targeted LAP requires accurate control of milk fermentation progress and LA production. This can be achieved through quantitative measurements and calibration plots using liquid chromatography measurements [18] and/or in situ nuclear magnetic resonance spectroscopy of the methyl protons [22]. The use of clay minerals displaying basicity and adsorption capacity via anion exchange such as Layered Double Hydroxides (LDH) can allow for overcoming the detrimental effect of excessive LA production [17].

Adsorption of non-dissociated LA molecules via organophilic interactions is also possible at low pH. However, around pH 4.3–4.7, competitive adsorption of non-charged caseins also takes place via similar interactions not only on LDH but also on high silica clay minerals. Both fermentation and coagulation in the presence of clay materials are strongly influenced by milk component interactions with the solid surface. This agrees with the dependence of fermentation on many factors, including the type of container surface [23]. In other words, fermentation and coagulation in stainless steel or aluminum tanks should differ from that performed in traditional clay-based pottery.

Acidic surfaces like those of ceramics or clay-based containers are expected to promote casein micelle destabilization and further coagulation upon medium acidification. The use of acidic surfaces must be similar to that already observed and demonstrated by the addition of highly silicic and acidic forms of montmorillonites [24]. Increasing pH is known to increase the temporary surface charge of all contacted species, making both fermentation and coagulation in the presence of clay materials dependent on the pH level, which in turn determines the distribution of the potential surface–milk interaction.

Surface interactions should also be involved in the less conventional electro-fermentation, which is intended to overcome the thermodynamic limitations of microbial fermentation. Nevertheless, this process was also found to favor detrimental hydrogen production through the so-called “dark fermentation” [25,26,27]. This must be due to the mere presence of a solid surface with promotes electron transfer will milk components. Nevertheless, the effect of the electrode materials remains to be elucidated. Unless the effect of the reactor’s internal walls is deeply investigated, the issues of protein stability and the unavoidable production of furfurals by other fermentative attempts, such as those performed upon Ohmic heating or ultrasonic treatment, still remain unexplained [28,29,30,31].

## 3. Shortcomings of Conventional Milk Clotting

Milk coagulation can also be triggered by chemical processes such as acidification in the presence of acidic species and/or proteolysis using a milk-clotting enzyme (MCE) [32]. The latter can be traditionally provided for most cheese manufacturers by different types of rennet, i.e., natural milk coagulant extracted from a specific part of the stomach of young mammal animals belonging to the ruminant family. Rennets and more particularly those extracted from gastric juice of young calves usually contain a Chymosin/Pepsin enzyme ratio higher than unity which, however, decreases down to 0.1–0.2 with aging. Both bovine and camel chymosin show enzymatic activity in bovine milk coagulation via casein destabilization.

Chymosin is a hydrolase-type enzyme and belongs to the peptidases family that act through proteolytic splitting of the phenylalanine–methionine bond of casein K (Phe_105_-Met_106_ bond) and phenylalanine–phenylalanine bond of casein Alpha using water molecules [33]. Camel chymosin (EC 3.4.23.4) turned out to be 2 up to 3-fold more active than its bovine counterpart in this process at pH 6.65, ionic strength 80 mM, but decreasing pH down to 6.00 and 5.50 induced an improvement of the enzymatic activity of both bovine and camel chymosins. Certain cheese manufacturers have also employed synthetic chymosin even though its use still remains a subject of controversy in the world and is forbidden in some European countries.

Most enzymes display proteolytic properties that lead to milk coagulation but do not necessarily produce stable coagulum. This is due to the occurrence of unstoppable proteolysis by enzymes such as trypsin and ficin and other proteolytic enzymes that even digest the very coagulum. Improvements in this regard consist in mitigating the activity of MCE [34].

Dairy technology often involves complex processes that depend on many parameters, which are difficult to simultaneously control during milk processing. Pronounced lactose fermentation into LA leads to excessive pH decrease that induces accentuated protein hydrolysis. The latter is known to affect the sensory properties of milk. In addition, large-scale dairy technology and more particularly cheese production often requires costly exploitation due to high MCE consumption. The high demand–supply ratio for calf rennet stimulated the search for viable and profitable sources of surrogates for traditional MCE [35,36]. Certain cheese manufacturers even employed synthetic chymosin even though its use still remains a subject of controversy in the world and is forbidden in some European countries [33].

## 4. Strategies for MCE Consumption Issue

For cheese production, the traditional strategy resides in MCE preparation by the original lactic bacteria-driven fermentation. Two other approaches involved other diverse fermentative routes (Table 1) and a wide variety of vegetal sources of enzymes (Table 2). Many other fungi and bacteria cultures originating from soils, butter, milk, and other sources were found to favor the formation of MCE displaying a wide range of clotting-to-proteolytic activity ratio of milk (MCA/PA) [37]. Other more or less successful attempts were carried out [38,39,40] more particularly on a variety of plant-deriving MCE in spite of their detrimental excessive proteolytic activity that confers bitter flavor to the manufactured cheeses [36]. In addition, in spite of their numerous advantages, the of use plant-deriving MCE still remains limited, due to potential food contamination risks by chemicals employed in the extraction process.

Other MCE for casein hydrolysis were isolated from micro-organisms such as *Bacillus licheniformis* BL312 [41], *Bacillus velezensis* DB219 [42] or *Mucor circinelloides* [43] and glutinous rice fermentation [44]. In all these works and others, the enzyme source was found to play a key role in the structure of the resulting αs-casein (αs-CN) and β-casein (β-CN) hydrolysates [45]. MCE such as cyprosin and cardosin can be prepared through the fermentation of some fungi like *Rhizomucor miehei*, *Rhizomucor pusillus*, *Mucor miehei*, *Aspergillus flavius*, *Aspergillus noir,* and others.

Enzyme performances are evaluated not only on their high milk-clotting activity (MCA) but also on their low proteolytic activity and MCA stability. The most suitable enzymes for profitable cheese manufacture should display the highest MCA/PA ratios and highest stability in time against pH and temperature fluctuations. Investigations are still in progress to improve both requirements.

**Table 1 ijms-25-00808-t001:** Some fermentative processes for milk-clotting enzyme preparation.

Micro-Organisms	Process	Enzyme Activity			Ref.
		MCA/PA *	Optimum Conditions	Stability Conditions	
*Thermomucor indicae-seudaticae N31*	Submerged fermentation 72 h/45 °C /150 rpm	MCA = 60.5 SU/mL MCA/PA= 510	pH 5.5/65 °C for MCA 60 °C for PA	pH 4.0–4.5/24 h; up to 55 °C for 1 h Storage: −20/+25 °C for 10 weeks	[46]
*Aspergillus oryzae DRDFS13*	Solid-state fermentation 30 °C/pH 6.0/5 days.	MCA = 137.58 SU/mL; MCA/PA = 1.30	Casein 0.5%/pH 4.0/25 °C,		[47]
*Penicillium roqueforti*	Cocoa shell fermentation	Highest at pH 10–12/80 °C	Addition of Na^+^, Co^2+^, Methanol, Ethanol, hexane	pH 10–12/80 °C.	[48]
*Thermomucor indicae-seudaticae N31*	Solid-state fermentation (SSF)	MCA inhibited by pepstatin A. Low PA	pH 5.7, at 70 °C and in 0.04 M CaCl_2_	pH 3.5–4.5/24 h; up to 45 °C for 1 h.	[49]
*Bacillus licheniformis BL312*	Fermentation substrate: 50 mM	MCA = 5291 SU/mg	CaCl_2_/pH 5.5/55 °C	pH 5.5–11.0; T ≤ 45 °C	[45]
*Bacillus velezensis DB219 from dairy soil (China)*	Fermentation wheat bran + soluble starch	3164.84 SU/mL MCA/PA = 9.2	Initial pH 6.15; 36 h/40 mL inoculum 5%		[42]
*Bacillus amyloliquefaciens JNU002*	At 48 h, 0.2% (*v*/*v*) inoculum; pH 6.0 in 15 L bioreactor	MCA = 4969 SU/mL PA = 4.02 SU/mL	MCA/PA = 2.575 at 35 °C, MCA/PA = 22.992) at 70 °C	pH 4–6; T ≤ 40 °C; No activity at 75 °C	[50]
*Bacillus subtilis B1, B. subtilis B18* and *B. thuringiensis B12*	Wheat bran fermentation	MCA B1 =131.50 B12 = 64.17 B18 = 114.09 SU/mL; MCA/PA = 4.75, 1.45, 3.55	pH 5.5; 50 mM CaCl_2_	Completely inactivated after 5 min at 70 °C	[51]
*Bacillus licheniformis 5A5*	Fermentation/static conditions	MCA = 21.9 SU/mL	37 °C for 48 h	73.4% rennet activity preserved/40 °C: 1 h	[52]
*Bacillus subtilis natto*	Fermentation at 175 rpm/1 day	MCA = 685 SU/mL; PA = 0.23	pH 6; 37 °C		[53]

* MCA: Milk-clotting activity; PA: Proteolytic activity. SU: Soxhlet units.

**Table 2 ijms-25-00808-t002:** Sources of enzymes for cheese production.

Vegetal Source	Used Fraction	Enzyme		MCA *	PA *	MCA/PA	Targeted Bond	Ref.
		Symbol	Nomenclature					
*Bromelia pinguin*	Plant fruit	CP, SP	B. pinguin extract	2.59 SU/mg	2.0 SU/mg	1.29	Phe_105_-Met_106_	[54]
*Calotropis gigantea*	Latex, stem, flower, and leaf	CP, SP	Calotropain	450 SU/mL	86.45 SU/mL	5.21	Phe_105_-Met_106_	[55]
*Solanum elaeagnifolium*	Fruit	-	Plant coagulant	4347.00 SU/mL	1.3 SU/mg	3343.00	Ser_104_-Phe_105_	[56]
*Cynara scolymus*	Flower	AP	Cynarase	147.65 SU/mg	5.45 SU/mg	27.1		[57,58]
*Ficus johannis*	Latex	CP		21.88 SU/mL	0.339 SU/mL	64.54		[59]
*Actinidia deliciosa*	Fruit	CP	Actinidin	2.7 SU/mg	0.55 SU/mg	5.00	Ala_90_-Glu_91_ His_102_-Leu_103_	[60]
*Cucumis melo*	Fruit	CP		1.5 SU/mg	0.90 SU/mg	2.5		[60]
*Zingiber officinale*	Fruit	CP	Ginger/Zingibain	2.3 SU/mg	0.73 SU/mg	3.2		[60]
*Morinda citrifolia*	Fruit	CP		238.8 SU/mL	8.86 SU/mg	27.00		[61]
*Balanites aegyptiaca*	Fruit	AP, SP	-	2.43 SU/mL	4.96 SU/mL	0.49		[62]
*Pergularia* *tomentosa*	Leaf, Latex	CP, SP	Enzymatic extracts	97.92 SU/mL 1246.45 SU/mL	24.66 SU/min 161.66 SU/min	3.96 7.70	Phe_105_-Met_106_	[63]

* Serine protease (SP), aspartic protease (AP), Cysteine protease (CP), milk-clotting activity (MCA), Proteolytic activity (PA).

## 5. Post-Process Strategies for Enzyme Recovery

Another strategy for addressing the issue of enzyme consumption resides in post-process enzyme recovery, i.e., enzyme separation after its use in a given process. Possible approaches in this regard involved enzyme immobilization on solids as already reported by the fairly extensive literature [64,65,66,67,68,69,70,71,72,73,74], and many works were devoted to food technology purposes [72,73,75,76,77,78,79,80,81]. The use of immobilized enzymes such as lactase for continuous hydrolysis of lactose in milk for the production of lactose-free milk is one of the most important applications in dairy technology [82].

So far, a wide variety of host matrices have been tested for enzymes [74]. These include various polymers, biopolymers (starch, cellulose, chitosan, chitin, collagen, cellulose, keratins, carrageenan, alginate), synthetic polymers (such as polyvinyl alcohols), bio-dendrimers (polyglycerols). In food technology, bone powder, anion exchange resins, agarose glutaraldehyde, and silica gel also turned out to be interesting materials to be envisaged for enzyme immobilization [83]. Enzyme immobilization procedures involve the formation of films or coatings. Transglutaminase enzymes were often used as crosslinking agents that modify the interaction with the impregnating media. For instance, crosslinking of gelatin-based films to reduce its hydrophilic interaction and, consequently, its solubility and biodegradability [84].

Most materials employed in enzyme immobilization display high hydrophilic character and beneficial dispersity in aqueous media. In some cases, when excessively dispersed in the form of fine particles, their solubility could be enhanced making their separation difficult to achieve. Possible improvements can be made in this regard by incorporating metal particles with magnetic properties [82,85]. In spite of potential contamination risks by metals and food safety constraints, an interesting attempt was achieved by the synthesis of Fe_2_O_3_-loaded chitosan microspheres for lactase immobilization (Table 3). This immobilized enzyme has the capacity to be easily recovered upon exposure to a magnetic field after milk processing [72]. Similar attempts allowed immobilizing β-galactosidase on commercially magnetite-doped chitosan gel [85]. Immobilized enzymes usually show higher stability towards pH change, thermal treatment, prolonged storage, and repeated reuse as compared to their free counterpart.

Enzyme immobilization usually takes place at the expense of the catalytic activity due to unavoidable interactions and/or chemical bonding of enzyme sites with the host matrices. Enzyme immobilization involves either purely physical adsorption or covalent bonding that unfortunately was found to affect the catalytic activity. However, it was also reported in attempts to improve low-fat cheese flavors that immobilized lipases on α-lactalbumin (α-lac) nanotubes (NTs) showed twice higher catalytic activity in the hydrolysis and release of free fatty acids (FFA) compared to free enzymes [79]. This result is of great importance because it clearly demonstrates that the type of host matrices determines the enzyme-matrice interactions and subsequently the catalytic activity in the targeted process.

The use of food-compatible biopolymers or bio-dendrimers like polyglycerols is of greater interest to the food industry [73,86,87]. However, most of these materials exhibit a major shortcoming that resides in their possible digestion upon fermentation. This drawback can be overcome by employing much more chemically stable inorganic materials such as mesoporous silicas [88] and ceramics [89]. Silicates and aluminosilicates such as zeolites and derivatives could also be interesting matrices for enzyme immobilization, in spite of their limited pore size. Among these, a special interest is devoted to clay materials, which are characterized by 2D lamellar structures. Like silicas [88], aluminosilicates such as cationic clay minerals bear variable surface charges and ion exchange capacity according to pH level, ionic force, and type of the clay–enzyme couple. Changes in these surface charges can significantly modify the dispersion or aggregation of enzyme-loaded matrices in the liquid media.

Montmorillonite, sepiolite, and bentonite were already used for the synthesis of new chitosan–clay composites for covalent immobilization of bromelain via crosslinking with glutaraldehyde [90]. This proteolytic enzyme, extracted from pineapple stems, was intended for winemaking processes. Here, unlike in other works, clay incorporation in the composite was found to improve the mechanical properties but slightly affected the catalytic activity. Fluctuations in enzyme activity from one clay material to another must be due to different compositions that promote different clay/enzyme interactions. Deep knowledge about the colloidal and pH-dependent behavior of clay materials is an essential requirement for avoiding non-conclusive results and sometimes contradictory performances.

Here, also, among numerous works on clay-immobilized enzymes reported so far, only very few were intended for the food industry [83,84] and much less for dairy technology purposes [91,92]. Even if the immobilization procedures are well established for various clay materials, there exists a glaring lock of data on the acid/base properties that govern the dispersion of clay-supported enzymes in the aqueous media [93].

**Table 3 ijms-25-00808-t003:** Use of immobilized enzymes in dairy technology.

Application in Dairy Technology	Enzyme	Matrice	Properties	Ref.
Lactose conversion into glucose and galactose	Lactase	Fe_3_O_4_-loaded chitosan microspheres	Magnetic feature for easy recovery. Improved stability towards pH change, thermal treatment, and prolonged storage.	[72]
Flavor improvement in low-fat cheeses via lipid hydrolysis and increase in free fatty acids release	Lipase	Oil–water emulsion of α-lactalbumin (α-lac) nanotubes (NTs)	Matrice formed by self-assembly of partially hydrolyzed α-lac peptides. Twice the amount of FFA released.	[79]
Lactose hydrolysis in milk		Freeze-dried capsules prepared from emulsions; solid/oil/water (S/O/W) emulsions	Spray-dried lactase powder was suspended in anhydrous milk fat/Span^®^ 80 emulsified by sodium caseinate and lecithin (5:1). The encapsulated lactase was released gradually during the simulated digestions to hydrolyze lactose in milk more efficiently than free lactase.	[94]
winemaking processes	Bromelain	Chitosan/clay nanocomposite films	Incorporation of montmorillonite, sepiolite, or bentonite in the composite improves the mechanical properties but slightly affects the catalytic activity.	[90]
Lactose-free milk manufacture for overcoming lactose intolerance	Lactase	Cellulose acetate	Film-based support for immobilization.	[95]
Lactose hydrolysis in milk		Freeze-dried capsules prepared from emulsions; solid/oil/water (S/O/W) emulsions	Spray-dried lactase powder was suspended in anhydrous milk fat/Span^®^ 80 emulsified by sodium caseinate and lecithin (5:1). The encapsulated lactase was released gradually during the simulated digestions to hydrolyze lactose in milk more efficiently than free lactase.	[94]
Lactose hydrolysis for lactose-free dairy products	Beta-galactosidase	Cellulose fiber	Adsorption/ionic on ion exchange modified cellulose fiber.	[96]
Batch coagulation of milk for feta-type cheese production	Chymosin (Rennin)	Cellulose/starch gel	Immobilized chymosin on a tubular cellulose/starch gel (TC/SG) composite.	[97]
Lactose-free milk production	Lactase	Cryogel disks	Cryogel disks prepared by free radical polymerization and chelated with Fe ions.	[92]

## 6. Lactic Ferment Inhibition and Lactic Acid Excess Issue

Lactic fermentation involves the conversion of lactose into LA and leads to the acidification of milk. This process can be inhibited by substances that reduce or block the action of lactic ferments. The inhibitors most frequently encountered in milk are the product of the lactose hydrolysis reaction itself, namely lactic acid, antibiotics resulting from medical treatment of the animal, pesticides (from contamination of livestock feed), preservatives, neutralizing agents often used in adulteration to reduce the acidity of milk and mask the reduced degree of freshness [1]. The presence of all potential inhibitors can be avoided by severe constraints in milk production, with the exception of lactic acid.

LA is a 2-hydroxypropionic acid that also results from anaerobic and O_2_-poor fermentation processes. In dairy technology, LA content in milk may be tailored according to the optimum sensory features of the targeted dairy products. This requires an equilibrium between milk acidification and the biologic activity of lactic bacteria (LB). This equilibrium can be altered by excessive LA fermentation. The resulting excessive acidity is known to affect not only the very growth of lactic acid bacteria (LAB) but also LAB activity [48,49,50,51,52,53], with negative impacts on the product quality and sensory properties [98,99,100,101]. These effects were already explained in terms of lactate anion protonation into non-dissociated LA molecules, more particularly around the p*K*_a_ (3.86). The latter can easily diffuse through bacterial cell walls. This unavoidably induces changes in the internal pH that affect the bacteria’s metabolisms [102,103]. LA was already reported to exhibit an inhibitory effect on bacteria growth and this effect is enhanced in combination with other micro-organisms and compounds present in the fermentation broth [104]. This inhibitory effect of increased LA concentrations on lactic bacteria cell growth was also observed during LA production for industrial applications.

LA concentration control during milk coagulation still remains a major challenge. Some strategies involved calf rennet substitution for skipping lactose fermentation into LA by using different MCE which resulted in different distribution of lactose derivatives. This unfortunately induced coagulum sensory properties different from those produced by traditional lactic fermentation. Other strategies devoted to various LA removal attempts by adsorption/anion exchange from milk fermentation broths have been reported so far [105,106,107,108,109,110,111,112,113,114,115,116,117,118,119].

Attempts to produce LA through conventional batch fermentation for industrial applications revealed that increasing LA concentration inhibits lactic bacteria cell growth and the very fermentation process. Advanced fermentation was also found to generate additional inhibitory species such as antibiotics, fungal metabolites, and amino acids [120]. In other LA production attempts by fermentation, electrodialysis turned out to be a more judicious removal procedure [121]. Continuous LA fermentation in the presence of strain *Lactobacillus paracasei* ATB 160111 revealed that only a certain part of the lactic bacteria still remains active and that the fermentative process is significantly hindered by inhibition products. Here, excessive lactic bacteria biomass upon advanced fermentation level was also found to paradoxically produce less LA [122]. The inhibitory capacity of LA was also demonstrated by the detrimental effect of its prolonged contact time with lactic bacteria in fluidized bed-based extractive fermentation in the presence of anion exchangers [123]. Extractive fermentation involving in situ ion exchange is an interesting technique that allows for overcoming the detrimental effect of LA. Nonetheless, most such attempts turned out quite unsatisfactory or cannot be applied to dairy technologies due to the use of synthetic materials as adsorbents or anion exchangers that cannot be compatible with food manufacturing constraints.

## 7. Continuous LA Removal from Fermentation Broths

So far, among the numerous attempts targeting direct LA removal from milk fermentation broths [104,124], most involve the use of weakly basic polymers or other resins as ion exchangers for lactate anion capture [123,125,126,127,128,129,130]. In this regard, Amberlite IRA400 resin showed promising performances with LA uptakes attaining 222.46 mg/g with easy consecutive LA release through mere elution in aqueous sulfuric acid (H_2_SO_4_), methanol, ammonia or their mixtures [127,131].

Other more or less successful attempts using Amberlite IRA900, IRA96, and IRA67 [131], revealed that weak base character of the anion exchanger is an essential requirement for effective LA removal. This is in agreement with the lower performances registered with a more acidic Amberlite IRA402 [132]. LA retention from fermentation broth using Amberlite IRA-400 anion exchanger was found to be of Langmuir type at pH 5.0, i.e., above the p*K*_a_ (3.86), with an almost twice higher retention capacity as compared to that of type II (multilayer adsorption) at pH 2.0 [127]. This confirms once again the detrimental effect of decreasing pH on LA adsorption. Continuous LA removal during fermentation using food-compatible adsorbents is a more viable approach [17,133,134]. This can be achieved by in situ LA retention in the very fermentation broth in batch mode (Figure 3).

Nonetheless, the limited adsorption capacity of most adsorbents imposes periodical alternate regeneration steps. This can be achieved by coupling the batch bioreactor to ex-situ adsorbent containing that allows LA desorption and elution in LA-free water or even LA mineralization into CO_2_ [18]. Aqueous suspension of anionic clay minerals such as Layered Double Hydroxides (LDH) or of their natural counterpart (hydrotalcite), which bears carbonate as a positive charge compensating anion can be used for lactate capture [7,135]. LDH ex situ regeneration can be achieved by ozone bubbling in the external aqueous media [18] or through back anion exchange with carbonate anion Figure 3).

LA removal for adsorbent regeneration by ozonation is an interesting route for such a purpose because if complete, this process should not generate harmful or toxic by-products. LA conversion into CO_2_ was already achieved by ozone in the absence [136,137] and the presence of solid catalysts [11,138,139,140,141,142,143]. Low-cost, non-toxic, and widely available food-compatible materials such as smectites (bentonite and derivatives) [11,138,139,140,141,142,143,144,145] and hydrotalcite and its combination with montmorillonite [18,136,137] already showed catalytic activity in such a process. In the regeneration of LA-loaded LDH by ozonation, the process was found to be enhanced when using additional cationic clay minerals that act as catalysts or co-catalysts [7]. Such materials belong to the smectite family, i.e., layered aluminosilicates (AS) bearing negative surface charges, usually compensated by food-compatible alkali or earth-alkaline cations [7]. In all cases, regardless of the adsorbents that may be in contact with milk fermentation broths, these materials should fulfill severe compatibility constraints for food technology purposes.

## 8. Clay-Based Materials as Food Supplement for Dairy Technology

Porous silicas and aluminosilicates, more particularly cationic clay materials, offer more than sufficient chemical stability for their uses in dairy technology. This has stimulated research in this direction, and so far, diverse attempts to use clay materials and clay-containing materials with direct or indirect interaction with milk production processing have been reported. Most of these attempts were related to the incorporation of clays such as bentonite and clay minerals like montmorillonite, sepiolite, and others as food supplements for milk cattle with more or less successful results. In spite of its fibrous structure [146], the European Union has already authorized the use of sepiolite as a binder-type feed additive (E562) in the category of technological additives in the nutrition of all animal species (EFSA, 2013) [147]. No sepiolite retention in the living organisms was observed, and the only safety issue was related to the nickel content that causes some inhalation risks as skin and respiratory sensitizer [148]. No specific study has been reported on the effect of the fiber length, but beyond the authorized 2% clay content in livestock feeding, sepiolite may be harmful like any other fibrous minerals such as asbestos.

Reportedly, the very clay consumption by animals was found to reduce the content of some harmful isotopes, most likely by adsorption, and to visibly improve milk production [149] and even to ease some health issues related to starch metabolization in cow hindgut [150]. These beneficial effects involve reduced glutamate dehydrogenase and γ-glutamyl transferase concentration, increased serum albumin and triglyceride production [151], and more balanced milk composition with noticeable enhancement in dairy milk production [2]. Among the toxins potentially present in milk, aflatoxins are particularly toxic and lethal for humans and animals, being able to block growth and cause cancer. There are various aflatoxins that differ by various chemical functions around an almost similar molecular polyaromatic core [152]. One aflatoxin is particularly of great interest, namely aflatoxin B_1_ (AFB_1_) is regarded as being a major issue in cattle nutrition and dairy technology. Cattle absorb AFB_1_ from mold-contaminated feed [153]. This is further metabolized via into aflatoxin M_1_ (AFM_1_) (Figure 4).

Given that the latter is excreted in milk, for safety reasons its maximum concentration of milk is restricted to 50 ng/kg [154]. This risk threshold varies throughout the world and should be put in context when dealing with potential cumulative effects on animal and human health. This explains why milk toxin capture is undoubtedly one of the most important benefits of clay mineral incorporation in livestock feed for dairy production.

## 9. Clay-Based Materials for Milk Toxin Capture

Notwithstanding that the use of organic compounds still remains subject to controversy for food technology, attempts at mycotoxin retention in aqueous media have already been performed on a nanofiber material modified by polydopamine and ionic liquids [155]. The reported work showed that toxin capture occurs via H-bridges, π-π interaction between carbon atoms along with electrostatic or hydrophobic interactions.

These multiple interactions arise from the very molecular structures of all functionalized organic compounds including mycotoxins and are also possible on much more food-compatible clay surfaces [155]. Clay materials are known to display sufficient thermal and chemical stability under the conditions of biological processes and technological milk processing. There exists ample literature related to the use of clay materials for milk toxin capture before and after lactation. Raw and modified bentonites [156,157,158,159] and smectites turned out to be the most investigated clay materials in aflatoxin retention. This is most likely due to their multiple interaction whose distribution is known to vary according to the chemical composition and pH. In most cases, the clay performance turned out to be more than satisfactory, as summarized in Table 4.

Montmorillonite and to a lesser extent kaolinite also showed promising adsorptive properties for antibiotics (oxytetracycline, chlortetracycline, tetracycline) at all pHs and low ionic strength [165]. This suggests that potential changes in the contributions of each type of interaction upon pH fluctuations do not significantly affect the adsorption capacity. The most plausible explanation resides in weak contributions of electrostatic adsorption and of usually preponderant hydrophobic interaction.

## 10. Hydrophilic–Hydrophobic Interactions of Clays in Aqueous Media

In aqueous media, dispersed clay materials behave as colloid-like suspensions [83]. Cationic clay minerals such as smectites are layered aluminosilicates (AS) with surfaces exhibiting negative electric charges commonly compensated by alkali or earth-alkaline cations. Their positively charged counterparts are anionic clay minerals that bear exchangeable anions (chloride, carbonate, etc.) [135]. They are so-called Layered Double Hydroxides (LDH), which have natural counterparts, namely hydrotalcites which differ according to their Mg/Al ratio and bear positive charges compensated by carbonate anions.

Both types of clay materials have already been tested in a series of applications including dairy technology. Indeed, the use of protonated montmorillonite (HMt) in milk acidification [134] and of LDH in LA removal [17] and total mineralization in harmless CO_2_ [18] are particularly interesting. Such attempts allowed for reducing not only the consumption of costly MCE and other lactic seeds but also the detrimental inhibitory effect of excessive LA production. These attempts provided evidence that clay surfaces simultaneously exhibit various interactions towards dispersed species and catalytic properties for the safe removal of harmful organic species. Smectite surfaces like montmorillonites are known to display electrostatic interactions due to the presence of cations and both permanent and temporary negative charges (cation-compensated sites and deprotonated silanol, respectively). The total negative charge is known to increase with increasing pH inducing an improvement of the cation exchange capacity. In addition, smectites also exhibit both hydrophilic behavior through their silanols and organophilic character induced by their siloxy groups (Si-O-Si).

Such interactions are intrinsically dependent on the chemical and mineralogical compositions of the clay material, more particularly by the global and smectite silica/alumina ratios and exchangeable cation. However, the contributions of some of these interactions may change upon fluctuations in the pH and chemical composition of the host liquid media, inducing significant modifications of the dispersion or coagulation–flocculation of the clay particles. High clay hydrophilic character accounts for high dispersion in aqueous media that leads to decreasing particle size. This has a direct effect on the extent of the contact surface accessible to other dispersed species.

Among these, organic molecules such as aflatoxins are expected to display high affinity towards clay surface as already confirmed by the increase in the retention capacity of aflatoxin M_1_ (AFM_1_) with decreasing particle size of raw bentonite [166]. On an isolated clay lamella, AFM_1_ molecules should adsorb via hydrophilic H-bridges between their -OH, -C=O, and even -C-O-C- groups with surface silanols. AFB_1_ adsorption is assumed to involve fewer H-bridges due to the lack of OH groups. In contrast, hydrophobic interactions should take place between their phenyl or methyl groups and siloxy islands on the clay surface (Figure 5).

Hydrophilic adsorption of AFM_1_ on smectite surfaces should expose the hydrophobic moiety towards the impregnating aqueous media (blue pathway). This is assumed to promote organophobic interaction, thereby favoring the adsorption of other AFM_1_ molecules and clay aggregation that reduces the accessible surface.

Conversely, AFM_1_ hydrophobic adsorption on surface siloxy groups improves the hydrophobic character of the surface by exposing the hydrophilic hydroxyls to the surrounding aqueous media (red pathway). This is expected to enhance the clay dispersion into finer particles and subsequently the surface adsorption capacity. Here, high silica clay surfaces are expected to promote ***i.*** primary hydrophobic interaction with organic molecules and ***ii.*** secondary hydrophilic interaction of the peripherical hydroxyl shell of the resulting organoclay with the surrounding aqueous media (Figure 6).

Silicic clay materials should be regarded as more suitable for the adsorption of organic molecules and more particularly aflatoxins. Therefore, the effects of high silica contents, low cation exchange capacity, and exchange with divalent cations having low hydration energy can explain, at least partly, the poorly understood role of the mineralogical and chemical properties of bentonites in the improvement of aflatoxin retention through hydrophobicity enhancement [167]. Nonetheless, highly hydrophobic character can also be detrimental due to the unavoidable retention of pharmaco-medicinal molecules and drugs such as antibiotics and organic nutrients (vitamins).

The different hydrophobicity of the four milk caseins can also induce selective adsorption and modifications in their equilibrium distribution, thus affecting the sensory features and nutritional value of milk itself. Low-silica or regular smectites are expected to exhibit the opposite phenomenon with external hydrophobic interaction shell (red) through their methyl groups oriented towards the aqueous media. This should result in a peripherical hydrophobic shell that attracts surrounding organoclay particles or aflatoxin molecules. The unavoidable clay particle aggregation leads to a decay of the external contact surface and of the adsorption capacity.

## 11. H-Bridges and Electrostatic Interactions

Reportedly, there exists a strong interdependence between the clay chemical composition and pH of the impregnating aqueous media since aluminum–iron-pillared montmorillonite (8.0 Al/Fe mole ratio) and smectite showed higher AFB_1_ adsorption capacity at alkaline to neutral pH than in acid media [163]. This must be due to the occurrence of electrostatic pH-dependent interactions, with possible ion exchange and surface protonation–deprotonation processes.

The incorporation of calcium montmorillonite (CaMt) in cattle food was found to reduce aflatoxin (AF) content without affecting milk production and composition [154]. Reportedly, AFB_1_ should adsorb via weak electrostatic interactions, moderate electron donor–acceptor coupling, and strong Ca^2+^ bridging linkage between the two C=O bonds of AF and the negative charge of the smectite surface [164,168]. This bridging effect of bivalent cation was already reported for a commercial smectite-type clay material in the presence of Ca^2+^ salts at alkaline pHs. This explains somehow the higher affinity of Ca^2+^/bentonite towards aflatoxin AFB_1_ at pH 2 and 6.5 as compared to Na^+^/bentonite [159].

At low pH, protonation processes are supposed to generate zwitterion forms of aflatoxins [169]. The hydroxyl group may also be protonated at very low pH by similarity with phenols. The latter can be protonated not only at their hydroxyls but also at certain positions of their aromatic rings when contacted with very strong superacids, depending on the acid strength [170]. This agrees with Zeta potential measurements which revealed that AF adsorbs via electrostatic interaction in some sections of the gastrointestinal (GI) tract [168].

Notwithstanding that silanol protonation into –SiOH_2_^+^ groups should take place only below the pH_pzc_ (pH of zero charges around 2–3) [159], aluminosilicates and more particularly bentonites are known to bear variable negative charges in the pH range 4–7 not only from the exchange sites but also from silanol deprotonation, i.e., in another pH range than reported by the literature [171]. These negative charges also promote electrostatic Clay: AFB_1_ interaction, but only weak contribution of AFB_1_ electrostatic adsorption on clay materials due to competitive AFB_1_ interaction with the GI membranes.

As previously stated, such interaction competitivity is assumed to vary according to the type and chemical composition of the clay materials. This result is of great importance because it clearly demonstrates that the use of high silica clay materials is more commended. This allows for avoiding dealumination, demetallization and other clay stability issues under severe chemical attacks and for achieving high AF uptakes from ingested cattle food via hydrophobic interaction. The same requirement should be applied to further excreted milk treatments, considering the absence of competitive GI interaction.

Similar results were reported for increased aflatoxin M1 affinity towards clay surfaces [172]. This result was somehow expected given that aluminosilicates generally display various types of interactions with surrounding chemical species. Since these interactions are pH-dependent, it clearly appears that mycotoxin adsorption should be different in the animal stomach at pH 3.5 and in the intestine transit at pH 6.5 [162]. Consequently, the clay materials used as food additives should display different features from those of their counterparts used in excreted milk treatment.

Direct use of clay materials in the purification of the already produced milk revealed appreciable retention capacity for hydrophobic saponite-rich bentonite and to a lesser extent, Kaolin. In spite of a slight alteration of the nutritional feature of the clay-treated milk, the aflatoxin content dropped from ca. 80 ng/L down to the maximum authorized concentrations of 50 ng/L for adults and 25 ng/L for lactants [173]. However, care should be taken for further fermentation of milk produced after clay-enriched food. Such livestock food can modify the microflora distribution in milk and the very targeted fermentation process as well. Indeed, Na^+^, K^+^, and Mg^2+^-exchanged bentonite samples were found to favor different abundances of various lactic bacteria strains in the production of some Asian fermented foods such as kimchi [174]. This result is of great importance because it clearly demonstrates that the mere presence of a solid surface and even the type of container and surface interaction with the liquid media can influence the fermentation process, in agreement with data related to the production of an African milk derivative traditionally prepared [23].

In some Mediterranean countries, traditional milk coagulation is still achieved in clay pottery vessels made from bentonite or kaolinite. These vessels are previously washed with culinary acids such as acetic and citric acids with or without seeding with a small amount of already clotted milk or sour creams. Such a procedure is assumed to lead to vessel surface protonation on exchangeable sites or even on surface silanols, which act as proton reservoirs for the coagulation step of fresh milk. Some of these vessels are coated by silica obtained by calcination of gel layers containing dissolved silicon-based species such as ortho-silicic acid. Silica is known to exhibit a hydrophobic character and two types of silanols (out of plane and in plane) that undergo enhanced deprotonation around pH 5.6 and 8.5, respectively [8].

## 12. Constraints in Clay Material Stability for Dairy Technology

Most clay materials are relatively stable when used both as cattle food complement and in milk processing. Chemical instability upon dealumination should trigger in excessively acidic media [10,12,14,15,16,175,176,177]. This is the case of the stomach which exhibits the lowest pH levels (2.0–3.8), unlike the gastrointestinal tract, esophagus, duodenum, and jejunum, which only display weak acidity.

Clays are not very soluble in HCl, but prolonged exposure to HCl affects the structure. For instance, montmorillonite structure alteration was found to take place at a 6 M HCl concentration, as illustrated by progressive IR intensity decay for the OH bending (930–800 cm^−1^) and Al–O–Si (524 cm^−1^) vibrations [13]. ^29^Si NMR analysis revealed that sepiolite and kaolinite acid attacks promote octahedral cation extraction resulting in layered porous silica [178]. In addition, pH decreases down to 2–3, i.e., close to the pH of zero-charge (pH_ZC_) of the silicic fraction of bentonites is supposed to cause progressive to total suppression of the surface charge and reduce the retention capacity restricting adsorption to hydrophobic interaction [159]. Chlorite turned out to be quite reactive to HCl mainly resulting in iron and aluminum release, and the acid attack appears to be influenced by the acid concentration and temperature [177].

Given the higher stability of silica layers in clay mineral structures, acid attack mainly involves the octahedral alumina layer by extracting trace to small amounts of alumina and Fe_2_O_3_ or MgO_2_ depending on the clay composition. This was already observed upon contacting strong acids with kaolinite and montmorillonite, which however showed different chemical resistance [179]. This phenomenon should be more pronounced for high-alumina materials such as kaolin and halloysite [180]. Their low chemical stability, reduced cation exchange capacity and weak hydrophobic character make them to be regarded as less effective in the capture of aflatoxins via hydrophobic interaction as compared to smectites and vermiculite. Nevertheless, Al-rich clay minerals display higher specific surface areas [181] that may compensate for these shortcomings by promoting electrostatic and other types of interactions with aflatoxins in certain pH ranges [164,168]. Among the different categories of clay minerals (lamellar, fibrous, and sesquioxide structures along with other minerals), fibrous materials such as allophanes and imogolites exhibit higher specific surface area but weaker conventional clay properties such as high crystallinity, swelling capacity, and plasticity, as compared to smectites and vermiculite [6,7,181]. Imogolite has a nanotubular structure with a 2 nm pore size that confers a higher specific surface area and subsequently increases adsorption capacity. Halloysite can hardly be classified due to a 1:1 layered structure like kaolinite but in a tubular form that differs from the layered structure of kaolinite. Halloysite also displays high specific surface area and increased adsorption capacity [182]. This feature is of great interest for aflatoxin retention in spite of its lower hydrophobic character.

Controlled acid activation can improve the hydrophilic character. Here, excessive removal of lattice iron or magnesium unavoidably induces a loss in surface charge but generates holes that improve the porosity and specific surface area. Rigorously controlled and moderate acid attacks are usually performed to achieve sufficient dealumination–demetallization targeting optimum hydrophobic character and porosity for improved toxin retention without producing structure crystallinity alteration and framework collapse [10,14,175].

## 13. Conclusions

The literature data analyzed herein show that some types of clay minerals, more particularly smectites, are suitable for some applications in dairy technology fulfilling all safety constraints imposed in both milk production and processing. Essential requirements for such purposes reside not only in deep knowledge of the physico-chemistry of clay materials and their interactions in aqueous with milk components and other species but also in multidisciplinary strategy. High silica clay minerals can even be regarded as harmless food-compatible materials in key processes such as fermentation and coagulation owing to their high chemical stability against acid attack. Clay interactions with dispersed species are key factors that govern adsorption and surface physicochemical processes in correlation with the pH, chemical composition of the liquid media, and, more importantly, clay chemical stability. Deep knowledge of clay material structures and interactions combined with a multidisciplinary approach are essential requirements for applications in dairy technology. This data analysis and synthesis provide valuable and useful findings for improving clay material behavior and performances and for designing green and safe processes in this regard.

## Data Availability

Not applicable.

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
