# Peer review of "Role of Clay Substrate Molecular Interactions in Some Dairy Technology Applications"

_ijms, 2024, doi:10.3390/ijms25020808_

Round 1

Reviewer 1 Report

Comments and Suggestions for Authors

1) The aim of the article should be explained at the end of the introduction section (Line 76)

2) There should be conclusion section to explain the further suggestions on this subject

3) If possible, the statistical data on clay consumption at dairy industry should be provided (Line 63)

4)Line 80. "Extensive literature..." . The citation to those litereture should be given at the end of the sentence

5)The legislation on use of clay materials as an ingredient or food additive should be explained

6) There are some recent articles on the application of clay material for dairy industry , therefore they should be included

Author Response

Reviewer #1:

Reviewer’s general comment

No general comment provided.

Authors’ response

All reviewer’s comments have been considered. point-by-point response to all these comments is herein attached to this revised version of the manuscript.

Reviewer’s specific comment

1) The aim of the article should be explained at the end of the introduction section (Line 76)

Authors’ response

Done.   This section has been improved with additional explanations and new supporting references. Two additional paragraphs have been added in the newly renumbered lines 77-104. See changes colored in green.

Reviewer’s specific comment

2) There should be conclusion section to explain the further suggestions on this subject

Authors’ response

Done.  A global conclusion section has been added at the end of the manuscript.

Reviewer’s specific comment

3) If possible, the statistical data on clay consumption at dairy industry should be provided (Line 63)

Authors’ response

Done. There is neither precise nor reliable global stats regarding world milk production and clay consumption, livestock feed production in dairy industry, because the use of such materials is still limited to at most livestock feeding, where mostly bentonites are used in Total Mix Ration (TMR) with 2% content in dietary dry matter. However, quick calculations for approximately 222 millions of milk animals throughout the world (about 133 million holdings keeping dairy cattle or 60 %; 28.5 million keeping buffaloes or 12.9 %; 41 and 19 million keeping goats and sheep, i.e. 18.5 % and 8.6 % respectively) with a hypothetically similar daily feed consumption of 50 pounds of dry matter (Food and Agriculture Organization of the United Nations, https://ourworldindata.org/grapher/milk-production-tonnes)

, and a maximum clay content of 2 % for high milk production [1, 2] give a maximum clay consumption of ca. 222 millions x 50 pounds x 0.454 kg/pd x 0.02 = 100 millions kg/day = 100 tons clay/day throughout the world.

To comply with this reviewer’s comment, only this estimated result of these calculations and some of these statements with two new references have been included in the revised manuscript.

Reviewer’s specific comment

4)Line 80. "Extensive literature..." . The citation to those litereture should be given at the end of the sentence

Authors’ response

Done. ‘’Extensive literature’’ means hundreds and even thousands of papers that cannot be cited herein, because they have nothing to do with the objectives of this present analysis paper. This sentence was only intended to show that clays materials have also been used in other applications (enzyme immobilization and lactic bacteria growth) than those considered in the present work.

However to comply with this reviewer’s comment, this sentence has been thoroughly rephrased as follows: ‘’ Numerous works have also been devoted in the forms of research and review papers along with technical and scientific reports to the applications of clay as adsorbents for the capture of various undesirable chemical species upstream (Aflatoxins) and downstream (fermentation inhibitors) of milk production‘’.

Reviewer’s specific comment

5)The legislation on use of clay materials as an ingredient or food additive should be explained

Authors’ response

Done.

So far, the EU has authorized 15 clay materials to be used as binder-type additives in animal nutrition [3] with a maximum clay content of 2 % for the production of high quantity and quality milk [2], Livestock feeds have been mostly enriched by bentonites and their deriving montmorillonites, alone or in mixtures diverse biocompatible organic compounds [4]. The authorized clay materials were considered as being safe for consumption since 2012 and non-genotoxic, non-skin irritants and non-skin sensitizer unless inhaled, since their silica content is hazard source for human and animal health [2]. Clay materials are also well-known exhibit sharp edge lamellae or pointed ends according to their structures, which provoke eye irritation by direct contact and even skin abrasion upon abusive cosmetic massages.

These statements have been added and marked in green color in lines 77-90 of the revised manuscript.

Reviewer’s specific comment

6) There are some recent articles on the application of clay material for dairy industry, therefore they should be included

Authors’ response

Done. Many new and recent references dealing with the reviewer’s comments have been added and cited in new statements in the revised manuscript. See below the list of the twelve added references

  1. Sulzberger, S.A., et al., Effects of clay after a grain challenge on milk composition and on ruminal, blood, and fecal pH in Holstein cows. Journal of Dairy Science, 2016. 99(10): p. 8028-8040.
  2. Rychen, G., et al., Efsa Panel on Additives Products or Substances used in Animal Feed -Safety and efficacy of bentonite as a feed additive for all animal species. EFSA Journal, 2017. 15(12): p. e05096.
  3. Nadziakiewicza, M., S. Kehoe, and P. Micek, Physico-Chemical Properties of Clay Minerals and Their Use as a Health Promoting Feed Additive. Animals (Basel), 2019. 9(10).
  4. Jiang, Y., et al. Aflatoxin in Dairy Cows: Toxicity, Occurrence in Feedstuffs and Milk and Dietary Mitigation Strategies. Toxins, 2021. 13, DOI: 10.3390/toxins13040283.
  5. Galan, E., Properties and applications of palygorskite-sepiolite clays. Clay Minerals, 1996. 31(4): p. 443-453.
  6. Burçak, E. and S. Yalçın, Effects of dietary sepiolite usage on performance, carcass characteristics, blood parameters and rumen fluid metabolites in Merino cross breed lambs. Applied Clay Science, 2018. 163: p. 291-298.
  7. Additives, E.P.o., et al., Safety and efficacy of a feed additive consisting of sepiolite for all animal species (Sepiol S.A and Tolsa, S.A). EFSA Journal, 2022. 20(4): p. e07250.
  8. Jeon, I. and K. Nam, Change in the site density and surface acidity of clay minerals by acid or alkali spills and its effect on pH buffering capacity. Scientific Reports, 2019. 9(1): p. 9878.
  9. Carroll, D. and H.C. Starkey, Reactivity of clay minerals with acids and alkalies. Clays and Clay Minerals, 1971. 19(5): p. 321-333.
  10. Neeraj, K. and M. Chandra, Basics of Clay Minerals and Their Characteristic Properties, in Clay and Clay Minerals, N. Gustavo Morari Do, Editor. 2021, IntechOpen: Rijeka. p. Ch. 2.
  11. Wang, S.-Y., et al., Occurrence of aflatoxins in water and decontamination strategies: A review. Water Research, 2023. 232: p. 119703.
  12. Zavala-Franco, A., et al., Assessing the Aflatoxin B₁ Adsorption Capacity between Biosorbents Using an In Vitro Multicompartmental Model Simulating the Dynamic Conditions in the Gastrointestinal Tract of Poultry. Toxins (Basel), 2018. 10(11).

Reviewer 2 Report

Comments and Suggestions for Authors

This work is a review about the use of clay based materials in diary applications with a particular focus on the clay-substrate interactions. Herein, the use of clay minerals for diary technology has origin in the food-compatible properties of these materials and it has the aim of improving some crucial processes such as fermentation or coagulation, also acting as adsorbent materials.

The experimental condition are also important and, among them, the pH, the chemical composition and the clay stability was focused.  Overall, this work is interesting but there are some major revisions that need to be addressed:

1.The use of clay based materials for actual diary technology should be reported and better discussed.

2.What is the effect in term of biocompatibility? After the treatment, the clay should be removed before ingestion or not? Some clays, for example sepiolite depending on the fiber length, can have harmful effects for the organisms. Please provide details.

3.The chemical and also the colloidal stability of clay is essential for this issue. The colloidal stability can be optimized by a pre-treatment of the clay in alkaline medium of acidic medium before being used with milk. The appearance of silanols groups can be useful. Please provide more details and paper for references about the treatement of clays in alkaline/acidic conditions, also focus on halloysite.

4.It would be interesting to divide clays depending on their morphology. Sheet-like clays should be less efficient in adsorbing molecules compared to, for examples, nanotubular clay. Please discuss better and also consider halloysite nanotubes in the manuscript as clay.

5.Please improve quality of images. Scheme 2 can be barely read.

Comments on the Quality of English Language

minor editing 

Author Response

Reviewer #2:

Reviewer’s general comment

Comments and Suggestions for Authors

This work is a review about the use of clay-based materials in diary applications with a particular focus on the clay-substrate interactions. Herein, the use of clay minerals for diary technology has origin in the food-compatible properties of these materials and it has the aim of improving some crucial processes such as fermentation or coagulation, also acting as adsorbent materials.

The experimental condition are also important and, among them, the pH, the chemical composition and the clay stability was focused.  Overall, this work is interesting but there are some major revisions that need to be addressed:

Authors’ response

Done.  

Done. All reviewer’s comments have been considered. point-by-point response to all these comments is herein provided in this revised version of the manuscript.

Reviewer’s specific comment

1.The use of clay based materials for actual diary technology should be reported and better discussed.

Authors’ response

Done. For this purpose, a series of sentences have been thoroughly reformulated including additional statements related to the use of clay materials for diary technology with supporting literature. See significant  changes marked in green color in the introduction section.

Reviewer’s specific comment

2.What is the effect in term of biocompatibility? After the treatment, the clay should be removed before ingestion or not? Some clays, for example sepiolite depending on the fiber length, can have harmful effects for the organisms. Please provide details.

Authors’ response

Done. Clay biocompatibility is mainly promoted by their chemical stability in biochemical media that host mostly fermentative process with acidity production. Some data on clay biocompatibility related to its chemical stability in moderately acidic aqueous media such as milk were already provided in:

  1. the 1st sentence of section 9. Clay-based materials as food supplement for dairy technology;
  2. the 2nd sentence of the paragraph before Table 4;
  3. the 5th paragraph of section 12. H-bridges and electrostatic interactions.
  4. the 2nd sentence of the 1st paragraph of section Constraints in clay material stability for dairy technology and supported by 6 references.
  5. the 1st sentence of the 3rd paragraph of section 13. Constraints in clay material stability for dairy technology

As already stated and supported by 6 references in section 13. Constraints in clay material stability for dairy technology, this stability is somehow limited by dealumination and demetallization risks in excessively acidic media of stomach which exhibits the lowest pH levels (2.0-3.8).

In addition, the authorized clay materials were already considered as being safe for consumption since 2012 and non-genotoxic, non-skin irritants and non-skin sensitizer. Nonetheless, their silica content is hazard source for human and animal health by inhalation [2]. Clay materials also exhibit sharp edge lamellae or pointed ends according to their structures, which provoke eye irritation by direct contact and even skin abrasion upon abusive cosmetic massages.

However, to comply with this reviewer’s comment, these last statements have been added and marked in green color in lines 77-90 of the revised manuscript.

The second part of this comment has been answered as a separate question. See below.

Reviewer’s specific comment

2…. After the treatment, the clay should be removed before ingestion or not? …

Authors’ response

Done.

When used in livestock feed, clay materials are ingested and eliminated with fecal wastes. In milk processing through fermentation and coagulation, practical considerations require the use of removable bars, plates or permeable bags containing clay materials in the form of grains or pellets, as already stated in the last sentence of the penultimate paragraph of section 2. Solid surface contribution in fermentation and coagulation in the submitted manuscript before scheme 2.

The third part of this comment has also been answered as a separate question. See below.

Reviewer’s specific comment

2…. Some clays, for example sepiolite depending on the fiber length, can have harmful effects for the organisms. Please provide details.

Authors’ response

Done.  In spite of its fibrous structure [5], the European Union has already authorised the use of sepiolite  as binder-type feed additive (E562) in the category of technological additives for all animal species (EFSA, 2013) in animal nutrition [6]. No safety issue was registered, since no sepiolite retention in the living organisms was observed, expect some risks upon inhalation as skin and respiratory sensitiser due to the nickel content [7]. No specific study has been reported on the effect of the fibre length, but beyond the authorized 2 % clay content in livestock feeding, sepiolite may be harmful like any other fibrous minerals such as asbestos.

Some of these statements have been reformulated and incorporated in lines 425-433 at the end of the 1st paragraph of section 9. Clay-based materials as food supplement for dairy technology.

Reviewer’s specific comment

3.The chemical and also the colloidal stability of clay is essential for this issue. The colloidal stability can be optimized by a pre-treatment of the clay in alkaline medium of acidic medium before being used with milk. The appearance of silanols groups can be useful. Please provide more details and paper for references about the treatement of clays in alkaline/acidic conditions, also focus on halloysite.

Authors’ response

Done.  Some statement related to the chemical stability are already provided as a response to the previous reviewer’s comment (See above).

The colloid behavior of clay materials in biochemical media was already examined in the 1st phrase of the penultimate paragraph of section 1. Introduction and in the last sentence of the penultimate paragraph of section 5. Post-process strategies for enzyme recovery, before Table 3.

However, since this question has been asked, additional statements with several supporting references have been incorporated in the revised manuscript to explain the colloidal and pH-dependent behavior of clay materials. See changes marked in green color in lines 94-104.

The second part of this comment related to ‘’ also focus on halloysite ‘’ is herein answered as a separate comment. See below.

Reviewer’s specific comment

3…. also focus on halloysite.

Authors’ response

Done. As already stated in section 13. Constraints in clay material stability for dairy technology of the manuscript (see the penultimate sentence of the penultimate paragraph before the newly added conclusion section), kaolin and halloysite have higher aluminum content and should display lower stability against acid attack as compared to montmorillonite [8, 9]. This  confers them weak hydrophobic character, and makes them to be regarded as less effective in the capture of aflatoxins via hydrophobic interaction as compared to smectites and vermiculite. Nevertheless, Al-rich clay minerals display higher specific surface areas [10] that may compensate these shortcoming by promoting electrostatic and other types of interactions with aflatoxins in certain pH ranges [11, 12].

Some of these statements have been rephrased and added in lines 638-644 in the last two paragraphs before the newly added conclusion section.

Reviewer’s specific comment

  1. It would be interesting to divide clays depending on their morphology. Sheet-like clays should be less efficient in adsorbing molecules compared to, for examples, nanotubular clay. Please discuss better and also consider halloysite nanotubes in the manuscript as clay.

Authors’ response

Done. It is well established that clay minerals are classified in lamellar, fibrous and sesquioxide structures and other minerals  Fibrous clay materials such as allophanes and imogolites exhibit weaker conventional clay properties such as high crystallinity, swelling capacity and plasticity as compared to smectites and vermiculite [10, 13, 14]. Imogolite has a nanotubular structure with 2 nm pore size that confers higher specific surface area and subsequently increased adsorption capacity. Halloysite can hardly be classified due to a 1:1 layer structure similar to that of kaolinite but in tubular form that differs from the layered structure of kaolinite particles. Halloysite also displays higher specific surface area and increased adsorption capacity of great interest for aflatoxin retention in spite of its lower hydrophobic character.

As requested, some of these statements have been incorporated in lines 650-660 in the penultimate paragraph before the conclusion section.

Reviewer’s specific comment

5.Please improve quality of images. Scheme 2 can be barely read.

Authors’ response

Done. Scheme 2 has been improved by increasing the characters size.

Reviewer’s specific comment

(x) Minor editing of English language required.

Authors’ response

Done. The text has been fully spelled and some grammatical revision has been made. The removed or modified grammatical errors and typos are marked in green color where corrected.

Reviewer 3 Report

Comments and Suggestions for Authors

Lines 1-76. This section does not add anything new or convincing. Make it more precise 

Line 136. Add a full stop at the end of the sentence

Line 145-148. Check for grammatical structure

Line 172. A major part of enzymes displays proteolytic properties...... Fix the sentence grammar

Lines 199-205, and table 1. Write bacterial and fungal species names according to the standard format

Table 1. Keep a uniform format for the temperature unit. Use space between the number and unit.

Line 219. Possibles approaches...... Fix the sentence structure error

Microspheres line 241, nanotubes line 252, and free fatty acids line 253, are hyperlinked. Better to add a reference here instead of hyperlinking. The same is required for Table 3

Lines 253-256. Fix the sentence structure 

Section 11. Hydrophilic and hydrophobic interactions.

Make it precise and remove unnecessary explanations.

Lines 514, 520 and 525. Is there any difference in AFB1 written in different forms? If no difference, then write AFB1 in uniform format.

Lines 584 and 592. Write HCl in the uniform standard format

At several points, after the given reference space is required.

A very lengthy review of the topic ends without any conclusive remarks and recommendations. It is better to add conclusions and recommendations based on the literature reviewed

The review can be further improved by writing precisely and removing unneeded explanations.

Comments on the Quality of English Language

Minor editing is required to fix sentence structure and grammar issues throughout the document.

Author Response

Reviewer #3:

Reviewer’s general comment

No general comment provided

Authors’ response

All reviewer’s comment have been considered. point-by-point response to all these comments is herein attached to this revised version of the manuscript.

Reviewer’s specific comment

(x) Minor editing of English language required.

Authors’ response

Done. The text has been fully spelled and some grammatical revision has been made. The removed or modified grammatical errors and typos are marked in green color where corrected.

Reviewer’s specific comment

Lines 1-76. This section does not add anything new or convincing. Make it more precise 

Authors’ response

Done.  Many phrases have been thoroughly rephrased and other statements with supporting literature have also been added. All changes are marked in green color in this location referred to by the reviewer.

Reviewer’s specific comment

Line 136. Add a full stop at the end of the sentence.

Authors’ response

Done. A dot has been added and marked in green color where missing.

Reviewer’s specific comment

Line 145-148. Check for grammatical structure

Authors’ response                                                                                                                                          

Done. The sentence referred to has been reformulated as follows: ‘’ Surface interactions should also be involved in the less conventional electro-fermentation, which is intended to overcome the thermodynamic limitations of microbial fermentation. Nevertheless, this process was also found to favor detrimental hydrogen production through the so-called ‘’dark fermentation’’ [15-17].  ‘’.

Reviewer’s specific comment

Line 172. A major part of enzymes displays proteolytic properties...... Fix the sentence grammar

Authors’ response

Done.  Two grammatical errors have been corrected and the sentence has been partly rephrased as follows: ‘’ Most enzymes display proteolytic properties that lead to milk coagulation but do not necessarily produce stable coagulum‘’.

Reviewer’s specific comment

Lines 199-205, and table 1. Write bacterial and fungal species names according to the standard format

Authors’ response

Done. All bacterial and fungal species names have been italicized. See changes marked in green color in the lines referred to and table 1.

Reviewer’s specific comment

Table 1. Keep a uniform format for the temperature unit. Use space between the number and unit.

Authors’ response

Done. This mistake has been corrected where need and some missing subscripts have been added in table1.

Reviewer’s specific comment

Line 219. Possibles approaches...... Fix the sentence structure error

Authors’ response

Done.  This sentence has been rephrased as follows: ‘’ Possibles approaches in this regard involved enzyme immobilization on solids as already reported by a fairly extensive literature [18-28], and many works were devoted to food technology purposes [26, 27, 29-35].‘’.

Reviewer’s specific comment

Microspheres line 241, nanotubes line 252, and free fatty acids line 253, are hyperlinked. Better to add a reference here instead of hyperlinking. The same is required for Table 3

Authors’ response

Done. All hyperlinks have been removed where occurring.  The changes made have marked in green color.

Reviewer’s specific comment

Lines 253-256. Fix the sentence structure 

Authors’ response

Done.   This sentence has been rephrased as follows: ‘’ This result is of great importance because it clearly demonstrates that the type of host-matrices determines the Enzyme-Matrice interactions and subsequently the catalytic activity in the targeted process ‘’.

Reviewer’s specific comment

Section 11. Hydrophilic and hydrophobic interactions.

Make it precise and remove unnecessary explanations.

Authors’ response

Done.  For the sake of clarity, some sentences have been thoroughly rewritten.

Reviewer’s specific comment

Lines 514, 520 and 525. Is there any difference in AFB1 written in different forms? If no difference, then write AFB1 in uniform format.

Authors’ response

Done.  All these different terms have been uniformized int ‘’ AFB1 ‘’

Reviewer’s specific comment

Lines 584 and 592. Write HCl in the uniform standard format

Authors’ response

Done.   HCl has been written uniformly where different.

Reviewer’s specific comment

At several points, after the given reference space is required.

Authors’ response

Done.  This has been corrected where need in a very few locations throughout the entire original manuscript. These errors were already removed in the templated version of the manuscript.

Reviewer’s specific comment

A very lengthy review of the topic ends without any conclusive remarks and recommendations. It is better to add conclusions and recommendations based on the literature reviewed

Authors’ response

Done. We did not intend to write a conclusion, because all the manuscript is a set of conclusions and, as a rule in many respectable review papers it is not recommended to insert a conclusion.

However, this question has been raised, a conclusion section has been added.

Reviewer’s specific comment

The review can be further improved by writing precisely and removing unneeded explanations.

Authors’ response

Done. Some basic knowledge details and explanation have been removed. Changes are marked in green color where made.

Comments on the Quality of English Language

Minor editing is required to fix sentence structure and grammar issues throughout the document.

Authors’ response

Done. The entire text has been fully spelled and some grammatical revision has been made. The removed or modified grammatical errors and typos are marked in green color where corrected.

Round 2

Reviewer 1 Report

Comments and Suggestions for Authors

All the comments has been taken şn to account. The paper can be accepted for publication

Reviewer 2 Report

Comments and Suggestions for Authors

The authors addressed all comments and suggestions. The manuscript can be accepted for publication. 

Reviewer 3 Report

Comments and Suggestions for Authors

NA